# The Influence of Temperature on Ozone Production under varying $NO_x$ Conditions – a modelling study

J. Coates[1], K. A. Mar[1], N. Ojha[2], and T. M. Butler[1]

[1]Institute for Advanced Sustainability Studies, Potsdam, Germany
[2]Atmospheric Chemistry Department, Max Planck Institute for Chemistry, Mainz, Germany

*Correspondence to:* J. Coates (jane.coates@iass-potsdam.de)

**Abstract.** Surface ozone is a secondary air pollutant produced during the atmospheric photochemical degradation of emitted volatile organic compounds (VOCs) in the presence of sunlight and nitrogen oxides ($NO_x$). Temperature directly influences ozone production through speeding up the rates of chemical reactions and increasing the emissions of VOCs, such as iso-prene, from vegetation. In this study, we used an idealised box model with different chemical mechanisms (MCMv3.2, CRIv2,
MOZART-4, RADM2, CB05) to examine the non-linear relationship between ozone, $NO_x$ and temperature, and compared this to previous observational studies. Under high-$NO_x$ conditions, an increase in ozone from 20 °C to 40 °C of up to 20 ppbv was due to faster reaction rates while increased isoprene emissions added up to a further 11 ppbv of ozone. The largest inter-mechanism differences were obtained at high temperatures and high-$NO_x$ emissions. CB05 and RADM2 simulated more $NO_x$-sensitive chemistry than MCMv3.2, CRIv2 and MOZART-4 which could lead to different mitigation strategies being
proposed depending on the chemical mechanism. The increased oxidation rate of emitted VOC with temperature controlled the rate of $O_x$ production, the net influence of peroxy nitrates increased net $O_x$ production per molecule of emitted VOC oxidised. The rate of increase in ozone mixing ratios with temperature from our box model simulations was about half the rate of increase in ozone with temperature observed over central Europe or simulated by a regional chemistry transport model. Modifying the box model setup to approximate stagnant meteorological conditions increased the rate of increase of ozone with
temperature as the accumulation of oxidants enhanced ozone production through the increased production of peroxy radicals from the secondary degradation of emitted VOCs. The box model simulations approximating stagnant conditions and the max-imal ozone production chemical regime reproduced the 2 ppbv increase in ozone per °C from the observational and regional model data over central Europe. The simulated ozone-temperature relationship was more sensitive to mixing than the choice of chemical mechanism. Our analysis suggests that reductions in $NO_x$ emissions would be required to offset the additional ozone
production due to an increase in temperature in the future.

## 1 Introduction

Surface-level ozone ($O_3$) is a secondary air pollutant formed during the photochemical degradation of volatile organic compounds (VOCs) in the presence of nitrogen oxides ($NO_x \equiv NO + NO_2$). Due to the photochemical nature of ozone production, it is strongly influenced by meteorological variables such as temperature (Jacob and Winner, 2009). In particular, heatwaves,

characterised by high temperatures and stagnant meteorological conditions, are correlated with high ozone levels as was the case during the European heatwave in 2003 (Solberg et al., 2008; Vautard et al., 2005). Furthermore, Otero et al. (2016) showed that temperature was a major meteorological driver of summertime ozone concentrations in many areas of central Europe.

Temperature primarily influences ozone production in two ways: speeding up the rates of many chemical reactions, and increasing emissions of VOCs from biogenic sources (BVOCs) (Sillman and Samson, 1995). While emissions of anthropogenic VOCs (AVOCs) are generally not dependent on temperature, evaporative emissions of some AVOCs do increase with temperature (Rubin et al., 2006). The review of Pusede et al. (2015) provides further details of the temperature-dependent processes impacting ozone production.

Regional modelling studies over the US (Sillman and Samson, 1995; Steiner et al., 2006; Dawson et al., 2007) examined the sensitivity of ozone production during a pollution episode to increased temperatures. These studies noted that increased temperatures (without changing VOC or $NO_x$-conditions) led to higher ozone levels, often exceeding local air quality guidelines. Sillman and Samson (1995) and Dawson et al. (2007) varied the temperature dependence of the PAN (peroxy acetyl nitrate) decomposition rate during simulations of the eastern US determining the sensitivity of ozone production with temperature to the PAN decomposition rate. In addition to the influence of PAN decomposition on ozone production, Steiner et al. (2006) correlated the increase in ozone mixing ratios with temperature over California to increased mixing ratios of formaldehyde, a secondary degradation production of many VOCs and an important radical source. Steiner et al. (2006) also noted increased emissions of BVOCs at higher temperatures in urban areas with high $NO_x$ emissions also increased ozone levels with temperature.

The modelling study of Vogel et al. (1999) looked at the $NO_y$ transition value as an indicator of ozone production sensitivity at different VOC and $NO_x$ conditions and investigated the sensitivity of this transition value with different meteorological conditions. Higher temperatures led to a higher transition value of $NO_y$ which was attributed to the faster thermal decomposition of PAN. Vogel et al. (1999) also showed vertical mixing and dry deposition decreased the transition value of $NO_y$ showing that ozone production is sensitive to other non-chemical processes.

Pusede et al. (2014) used an analytical model constrained by observations over the San Joaquin Valley, California to infer a non-linear relationship between ozone, temperature and $NO_x$, similar to the well-known non-linear relationship of ozone production on $NO_x$ and VOC levels (Sillman, 1999). Moreover, Pusede et al. (2014) showed that temperature can be used as a surrogate for VOC levels when considering the relationship of ozone under different $NO_x$ conditions.

Environmental chamber studies have also been used to analyse the relationship of ozone with temperature using a fixed mixture of VOCs. The chamber experiments of Carter et al. (1979) and Hatakeyama et al. (1991) showed increases in ozone from a VOC mix with temperature. Both studies compared the concentration time series of ozone and nitrogen-containing compounds ($NO_x$, PAN, $HNO_3$) at various temperatures linking the maximum ozone concentration to the decrease in PAN concentrations at temperatures greater than 303 K.

The review of Pusede et al. (2015) highlights a general lack of modelling studies looking at the relationship of ozone with temperature under different $NO_x$ conditions. The regional modelling studies described previously concentrated on reproducing ozone levels (using a single chemical mechanism) over regions with known meteorology and $NO_x$ conditions then varying the

temperature. These regional modelling studies did not consider the relationship between ozone, $NO_x$ and temperature. Vogel et al. (1999) only considered the effect of faster reaction rates at higher temperature and not the additional contribution of increased biogenic emissions to ozone levels at higher temperatures.

Comparisons of different chemical mechanisms, such as Emmerson and Evans (2009) and Coates and Butler (2015), showed that different representations of tropospheric chemistry influenced ozone production. Neither of these studies examined the ozone-temperature relationship differences between chemical mechanisms. Furthermore, Rasmussen et al. (2013) acknowledged that the modelled ozone-temperature relationship may be sensitive to the choice of chemical mechanism and recommended investigating this sensitivity. Comparing the ozone-temperature relationship predicted by different chemical mechanisms is potentially important for modelling of future air quality due to the expected increase in heatwaves (Karl and Trenberth, 2003).

In this study, we use an idealised box model to determine how ozone levels vary with temperature under different $NO_x$ conditions. We determine whether faster chemical reaction rates or increased BVOC emissions have a greater influence on instantaneous ozone production with higher temperature under different $NO_x$ conditions. Furthermore, we compare the ozone-temperature relationship produced by different chemical mechanisms and determine which chemical processes drive the increase of ozone with temperature. Finally, we compare the rate of increase of ozone with temperature obtained from the box model to both observations and regional model ouput and consider the role of stagnation on the rate of increase of ozone with temperature.

## 2 Methodology

### 2.1 Model Setup

We performed idealised simulations using the MECCA box model (Sander et al., 2005) to determine the important gas-phase chemical processes for ozone production under different temperatures and $NO_x$ conditions. The MECCA box model was set up as described in Coates and Butler (2015) and updated to include vertical mixing with the free troposphere using a diurnal cycle for the PBL height. The vertical mixing scheme was based on the approach of Lourens et al. (2016) with the model using the mean mixing layer height from the BAERLIN campaign over Berlin, Germany (Bonn et al., 2016).

Our simulations were designed as an idealised case and not to be exact representations of any particular place. The simulations used a latitude of $51\,°N$, broadly representative of conditions in central Europe, and were run for daylight hours in one full day. Methane was fixed at $1.7$ ppmv throughout the model run, carbon monoxide (CO) and ozone were initialised at $200$ ppbv and $40$ ppbv and then allowed to evolve freely throughout the simulation. All VOC emissions were held constant until noon simulating a plume of freshly-emitted VOC. the mixing ratios of $O_3$, CO and $CH_4$ in the free troposphere were respectively set to $50$ ppbv, $116$ ppbv and $1.8$ ppmv. These conditions were taken from the MATCH-MPIC chemical weather forecast model on the 21st March (the start date of the simulations). The model results (http://cwf.iass-potsdam.de/) at the $700$ hPa height were chosen and the daily average was used as input into the boxmodel.

Separate box model simulations were performed by systematically varying the temperature between 288 and 313 K (15–40 °C) in steps of 0.5 K. NO emissions were systematically varied between $5.0 \times 10^9$ and $1.5 \times 10^{12}$ molecules(NO) cm$^{-2}$ s$^{-1}$ in steps of $1 \times 10^{10}$ molecules(NO) cm$^{-2}$ s$^{-1}$ at each temperature step. At 20 °C, these NO emissions corresponded to peak NO$_x$ mixing ratios of 0.02 ppbv and 10 ppbv respectively, this range of NO$_x$ mixing ratios covers the NO$_x$ conditions found in pristine and urban conditions (von Schneidemesser et al., 2015).

All simulations were repeated using different chemical mechanisms to investigate whether the relationship between ozone, temperature and NO$_x$ changes using different representations of ozone production chemistry. The reference chemical mechanism was the near-explicit Master Chemical Mechanism, MCMv3.2, (Jenkin et al., 1997, 2003; Saunders et al., 2003; Rickard et al., 2015). The reduced chemical mechanisms in our study were Common Representative Intermediates, CRIv2 (Jenkin et al., 2008), Model for OZone and Related Chemical Tracers, MOZART-4 (Emmons et al., 2010), Regional Acid Deposition Model, RADM2 (Stockwell et al., 1990) and the Carbon Bond Mechanism, CB05 (Yarwood et al., 2005). These reduced chemical mechanisms were chosen as they are all currently used by modelling groups in 3D regional and global models (Baklanov et al., 2014). Coates and Butler (2015) described the implementation of these chemical mechanisms in MECCA.

The chemical mechanisms use temperature-dependent rate constants, k(T), to represent temperature-dependent chemical processes, including the initial oxidation of VOC, peroxy nitrate (RO$_2$NO$_2$) formation and destruction, and reactions between peroxy radicals and NO leading to alkyl nitrate (RONO$_2$) formation. However, not all chemical mechanisms represent the same chemical processes by a temperature-dependent rate constant. For example, in CB05, the rate constant of RONO$_2$ formation during isoprene degradation is temperature dependent while RONO$_2$ formation during alkane degradation is temperature independent. Furthermore, none of the chemical mechanisms in our study represent the RONO$_2$ branching ratio as a temperature dependent process. Laboratory experiments have shown the temperature dependence of the RONO$_2$ branching ratio for some VOCs (Atkinson et al., 1987) but generally RONO$_2$ chemistry is not well known (Pusede et al., 2015) and this level of detail is not represented by the chemical mechanisms. Before chemical mechanisms can include the temperature-dependence of the RONO$_2$ branching ratio, further research is required.

Model runs were repeated using a temperature-independent and temperature-dependent source of BVOC emissions to determine the relative importance of increased emissions of BVOC and faster reaction rates of chemical processes for the increase of ozone with temperature. Many types of VOCs are emitted from vegetation with isoprene and monoterpenes globally having the largest emissions, 535 and 162 Tg yr$^{-1}$ respectively (Guenther et al., 2012). Temperature-dependent emissions of these highly-reactive BVOC in urban areas during the summer months have been linked to high levels of ozone pollution. For example, Wang et al. (2013) attributed high summertime levels of ozone in Taipei to increased isoprene emissions from vegetation during the hotter summer months. Vegetation in urban areas also provides additional ozone sinks through stomatal uptake and ozonolysis of emitted BVOCs, the review of Calfapietra et al. (2013) discusses the role of BVOCs emitted by trees in urban areas in more detail.

Biogenic emissions of monoterpenes and isoprene are included in all model simulations. Model runs using a temperature-dependent source of BVOC emissions considered only the temperature-dependence of isoprene emissions as specified by MEGAN2.1 (Guenther et al., 2012), Sect. 2.3 provides further details. Since isoprene is the most important BVOC on the

global scale, we focused on the influence of the temperature-dependent biogenic emissions of isoprene on ozone levels. Future work should assess the influence of temperature-dependent biogenic emissions of monoterpenes on ozone production. In the temperature-dependent set of model simulations, only isoprene emissions were dependent on temperature and all other emissions were constant in all simulations. In reality, evaporative emissions from anthropogenic sources increase with temperature (Rubin et al., 2006) and isoprene has also been measured from vehicular exhausts (Borbon et al., 2001). Representing a temperature-dependent evaporative source of AVOC and an anthropogenic source of isoprene requires detailed local knowledge of these emission sources (such as the traffic fleet). Since our box modelling study was designed as an idealised study and not to characterise the influence of all temperature-dependent emission sources in a particular region, we have not considered the potentially larger increase of ozone at higher temperatures due to these additional emission sources. Further modelling work assessing the influence of these temperature-dependent emission sources on ozone production would be useful for mitigating ozone pollution in urban areas.

Simulations were also performed to assess the role of mixing on the increase of ozone with temperature. In these box model simulations, the box model was set up as described previously but without mixing of the chemical species with the free troposphere. Thus, these simulations approximate stagnant conditions that favour accumulation of secondary VOC oxidation products and enhanced ozone production.

## 2.2 VOC Emissions

Emissions of urban AVOC over central Europe were taken from the TNO-MACC_III emission inventory for the Benelux (Belgium, Netherlands and Luxembourg) region for the year 2011. TNO-MACC_III is the updated TNO-MACC_II emission inventory created using the same methodology as Kuenen et al. (2014) and based upon improvements to the existing emission inventory during AQMEII-2 (Pouliot et al., 2015).

Temperature-independent emissions of isoprene and monoterpenes from biogenic sources were calculated as a fraction of the total AVOC emissions from each country in the Benelux region. This data was obtained from the supplementary data available from the EMEP (European Monitoring and Evaluation Programme) model (Simpson et al., 2012). Temperature-dependent emissions of isoprene are described in Sect. 2.3.

Table 1 shows the quantity of VOC emissions from each source category and the temperature-independent BVOC emissions. These AVOC emissions were assigned to chemical species and groups based on the profiles provided by TNO. The NMVOC emissions were speciated to MCMv3.2 species as described by von Schneidemesser et al. (2016). For simulations done with other chemical mechanisms, the VOC emissions represented by the MCMv3.2 were mapped to the mechanism species representing VOC emissions in each reduced chemical mechanism based on the recommendations of the source literature and Carter (2015). The VOC emissions in the reduced chemical mechanisms were weighted by the carbon numbers of the MCMv3.2 species and the emitted mechanism species, thus keeping the amount of emitted reactive carbon constant between simulations. The supplementary data outlines the primary VOC and calculated emissions with each chemical mechanism.

## 2.3 Temperature Dependent Isoprene Emissions

Temperature-dependent emissions of isoprene were estimated using the MEGAN2.1 algorithm for calculating the emissions of VOC from vegetation (Guenther et al., 2012). Emissions from nature are dependent on many variables including temperature, radiation and age of vegetation but for the purpose of our study all variables except temperature were held constant. The aim of the study was to determine the additional influence of temperature-dependent isoprene emissions on top of the temperature-dependent chemistry. In order to achieve this aim, we chose the MEGAN2.1 parameters used to calculate isoprene emissions online by the model to give similar isoprene mixing ratios at 20 °C to the temperature-independent emissions of isoprene. MEGAN2.1 was used to reflect the temperature-dependent emission profile of isoprene emissions and not to accurately represent the isoprene emissions of a particular region. The estimated emissions of isoprene with MEGAN2.1 using these assumptions are illustrated in Fig. 1 and show the expected exponential increase in isoprene emissions with temperature (Guenther et al., 2006).

The estimated emissions of isoprene at 20 °C lead to 0.07 ppbv of isoprene in our simulations while at 30 °C, the increased emissions of isoprene using MEGAN2.1 estimations lead to 0.35 ppbv of isoprene in the model. A measurement campaign over Essen, Germany (Wagner and Kuttler, 2014) measured 0.1 ppbv of isoprene at temperature 20 °C and 0.3 ppbv of isoprene were measured at 30 °C. The similarity of the simulated and observed isoprene mixing ratios indicates that the MEGAN2.1 variables chosen for calculating the temperature-dependent emissions of isoprene were suitable for simulating urban conditions over central Europe.

## 3 Results and Discussion

### 3.1 Relationship between Ozone, $NO_x$ and Temperature

Figure 2a depicts the contours of peak mixing ratio of ozone from each simulation as a function of the total $NO_x$ emissions and temperature when using a temperature-independent and temperature-dependent source of isoprene emissions for each chemical mechanism. The relative difference in ozone mixing ratios produced using each chemical mechanism from the MCMv3.2 is shown in Fig. 2b. A non-linear relationship of ozone mixing ratios with $NO_x$ and temperature is produced by each chemical mechanism. This non-linear relationship is similar to that determined by Pusede et al. (2014) using an analytical model constrained to observational measurements over the San Joaquin Valley, California.

Higher peak ozone mixing ratios are produced when using a temperature-dependent source of isoprene emissions (Fig. 2a). The highest mixing ratios of peak ozone are produced at high temperatures and moderate emissions of $NO_x$ regardless of the temperature dependence of isoprene emissions. Conversely, the least amount of peak ozone is produced with low emissions of $NO_x$ over the whole temperature range $(15 - 40 \,°C)$ when using both a temperature-independent and temperature-dependent source of isoprene emissions. The larger increases in ozone levels in the Maximal-O3 and High-$NO_x$ regimes indicate that strong reductions in $NO_x$ emissions are neccessary to offset the increase in ozone pollution at higher temperatures, especially in urban areas containing a significant amount of isoprene emitting vegetation.

As shown in Fig. 2b, regions of high temperatures and high $NO_x$ emissions generally lead to the largest inter-mechanism differences between ozone mixing ratios using reduced chemical mechanisms from the MCMv3.2 (up to 13 %). These differences in peak ozone mixing ratio produced from the reduced chemical mechanisms compared with the MCMv3.2 in each $NO_x$ condition are consistent with Fig. 3 (described below) where RADM2 and CB05 generally produced higher ozone levels than the MCMv3.2. Also consistent with Fig. 3, CRIv2 produced the most similar amounts of ozone to the MCMv3.2 in each $NO_x$ condition whereas MOZART-4 tended to produce lower ozone mixing ratios than the MCMv3.2 in High-$NO_x$ conditions. In Fig. 3, a maximum difference of 10 ppbv between ozone mixing ratios produced using the chemical mechanisms is reached at 40 °C in the High-$NO_x$ state when using both a temperature-independent and temperature-dependent source of isoprene emissions.

The $NO_x$ emissions required for maximum ozone production (the contour ridges in Fig. 2a) at each temperature is displayed in Fig. 1 of the supplementary material. This figure illustrates that RADM2 and CB05 require higher $NO_x$ emissions than the MCMv3.2 to achieve maximum ozone production at each temperature for both a temperature-independent and temperature-dependent source of isoprene emissions. At 20 °C, maximum ozone production is reached with $\sim 30$ % more $NO_x$ emissions using CB05 and RADM2 than the MCMv3.2 with a temperature-independent and temperature-dependent source of isoprene emissions. The CRIv2 and MOZART-4 chemical mechanisms require very similar $NO_x$ emissions to the MCMv3.2 at each temperature to produce maximum levels of ozone. Thus when modelling the air quality over a particular region using RADM2 and CB05, these mechanisms would be expected to simulate more $NO_x$-sensitive chemistry and a lower increase of ozone with temperature than the MCMv3.2, CRIv2 and MOZART-4 chemical mechanisms for the same conditions (i.e. emissions, meteorology and radiation).

The contours of ozone mixing ratios in Fig. 2a as a function of $NO_x$ and temperature can be split into three $NO_x$ regimes (Low-$NO_x$, Maximal-$O_3$ and High-$NO_x$), similar to the $NO_x$ regimes defined for the non-linear relationship of ozone with VOC and $NO_x$. The Low-$NO_x$ regime corresponds with regions having little increase in ozone with temperature, also called the $NO_x$-sensitive regime. The High-$NO_x$ (or $NO_x$-saturated) regime is when ozone levels increase rapidly with temperature. The contour ridges correspond to regions of maximal ozone production; this is the Maximal-$O_3$ regime. Pusede et al. (2014) showed that temperature can be used as a proxy for VOC, thus we assigned the ozone mixing ratios from each box model simulation to a $NO_x$ regime based on the $H_2O_2$:$HNO_3$ ratio. This ratio was used by Sillman (1995) and Staffelbach et al. (1997) to designate ozone to $NO_x$ regimes based on $NO_x$ and VOC levels. The Low-$NO_x$ regime corresponds to $H_2O_2$:$HNO_3$ ratios less than 0.5, the High-$NO_x$ regime corresponds to ratios larger than 0.3 and ratios between 0.3 and 0.5 correspond to the Maximal-$O_3$ regime.

The peak ozone mixing ratio from each simulation was assigned to a $NO_x$ regime based on the $H_2O_2$:$HNO_3$ ratio of that simulation. The peak ozone mixing ratios assigned to each $NO_x$ regime at each temperature were averaged, and illustrated in Fig. 3 for each chemical mechanism and each type of isoprene emissions (temperature independent and temperature dependent). We define the absolute increase in ozone from 20 °C to 40 °C due to faster reaction rates as the difference between ozone mixing ratios from 20 °C to 40 °C when using a temperature-independent source of isoprene emissions. When using a temperature-dependent source of isoprene emissions, the difference in ozone mixing ratios from 20 °C to 40 °C minus the increase due to

faster reaction rates, gives the absolute increase in ozone mixing ratios from increased isoprene emissions. These differences are represented graphically in Fig. 3 and summarised in Table 2.

Table 2 shows that the absolute increase in ozone with temperature due to chemistry (i.e. faster reaction rates) is larger than the absolute increase in ozone due to increased isoprene emissions for each chemical mechanism and each $NO_x$ regime. In all cases the absolute increase in ozone with temperature is largest under High-$NO_x$ conditions and lowest with Low-$NO_x$ conditions (Fig. 3 and Table 2). The increase in ozone mixing ratio from 20 °C to 40 °C due to faster reaction rates with High-$NO_x$ conditions is almost double that with Low-$NO_x$ conditions. In the Low-$NO_x$ regime, the increase of ozone with temperature using the reduced chemical mechanisms (CRIv2, MOZART-4, CB05 and RADM2) is similar to that from the MCMv3.2. Larger differences occur in the Maximal-$O_3$ and High-$NO_x$ regimes.

All reduced chemical mechanisms except RADM2 have similar increases in ozone due to increased isoprene emissions as the MCMv3.2 (Table 2). RADM2 produces 3 ppbv less ozone than the MCMv3.2 due to increased isoprene emissions in each $NO_x$ regime, indicating that this difference is due the representation of isoprene degradation chemistry in RADM2.

Coates and Butler (2015) compared ozone production in different chemical mechanisms to the MCMv3.2 using the TOPP metric (Tagged Ozone Production Potential) as defined in Butler et al. (2011) and showed that less ozone is produced per molecule of isoprene emitted using RADM2 than with MCMv3.2. The degradation of isoprene has been extensively studied and it is well-known that methyl vinyl ketone (MVK) and methacrolein are signatures of isoprene degradation (Atkinson, 2000). All chemical mechanisms in our study except RADM2 explicitly represent MVK and methacrolein (or in the case of CB05, a lumped species representing both these secondary degradation products). RADM2 does not represent methacrolein and the mechanism species representing ketones (KET) is a mixture of acetone and methyl ethyl ketone (MEK) (Stockwell et al., 1990). Thus the secondary degradation of isoprene in RADM2 is unable to represent the ozone production from the further degradation of the signature secondary degradation products of isoprene, MVK and methacrolein. Updated versions of RADM2, RACM (Stockwell et al., 1997) and RACM2 (Goliff et al., 2013), sequentially included methacrolein and MVK and with these updates the ozone production from isoprene oxidation approached that of the MCMv3.2 (Coates and Butler, 2015).

Our simulations produced a non-linear relationship between ozone, temperature and $NO_x$ with the absolute increase in ozone with temperature due to temperature-dependent chemistry larger than the increase in ozone with temperature due to temperature-dependent isoprene emissions. These results are consistent between each chemical mechanism, although for the same $NO_x$ and VOC conditions RADM2 and CB05 simulate a more $NO_x$-sensitive regime at the same temperature than the other chemical mechanisms (MCMv3.2, CRIv2, MOZART-4). In order to determine the chemical processes responsible for the increased ozone with temperature, we analyse the production and consumption budgets of ozone in Sect. 3.2.

## 3.2 Ozone Production and Consumption Budgets

Since chemical reactions contributing to both production and consumption of $O_x$ ($\equiv O_3 + NO_2 + O(^1D) + O$) have temperature-dependent rate constants, we analysed the production and consumption budgets of $O_x$ to determine the temperature-dependent chemical processes controlling the increase of ozone with temperature which was shown in Fig. 3. The $O_x$ budgets displayed

in Fig. 4 are assigned to each $NO_x$ regime for each chemical mechanism and source of isoprene emissions. The net production or consumption of $O_x$ is also indicated in Fig. 4.

Figure 4 was obtained by determining the chemical reactions producing and consuming $O_x$ and then allocating these reactions to important categories. Reactions of peroxy radicals with NO produce $O_x$ and the peroxy radicals are divided into 'HO2', 'RO2', 'ARO2' categories representing the reactions of NO with $HO_2$, alkyl peroxy radicals and acyl peroxy radicals respectively. Thus at each time step the $O_x$ production rate is given by

$$k_{HO_2+NO}[HO_2][NO] + \sum_i k_{RO_2,i+NO}[RO_{2,i}][NO] + \sum_j k_{ARO_2,j+NO}[ARO_{2,j}][NO] \tag{1}$$

for each alkyl peroxy radical $i$ and acyl peroxy radical $j$. The net contributions of peroxy nitrates, inorganic reactions and any other remaining organic reactions to the $O_x$ budget are represented by the 'RO2NO2', 'Inorganic' and 'Other Organic' categories in Fig. 4. The net contributions of these categories to the $O_x$ budget was calculated by subtracting the consumption rate from the production rate of the reactions contributing to each category. For example, peroxy nitrates produce $O_x$ when thermally decomposing or reacting with OH and consume $O_x$ when produced. Hence, at each time step the net contribution of RO2NO2 to the $O_x$ budget was calculated by

$$\sum_k k_{RO_2NO_2,k}[RO_2NO_{2,k}] + \sum_k k_{RO_2NO_2,k+OH}[RO_2NO_{2,k}][OH] - \sum_k k_{RO_2,k+NO_2}[RO_{2,k}][NO_2] \tag{2}$$

for each peroxy nitrate species $k$. The cumulative day-time budgets were calculated by summing the net contributions of the reaction rates of each category over the day-time period. The ratio of net ozone to net $O_x$ production was practically constant with temperature in all cases showing that using $O_x$ budgets as a proxy for ozone budgets was suitable at each temperature in our study.

The absolute production and consumption budgets allocated to the major categories are displayed in Fig. 4a. Both production and consumption of $O_x$ increase with temperature for each chemical mechanism and each $NO_x$ conditions. The overall net increase of $O_x$ production with temperature (white line in Fig. 4a) is consistent with the increase in ozone mixing ratios for each panel in Fig. 3. Moreover, the net chemical production of $O_x$ is larger when using a temperature-dependent source of isoprene emissions, again this is consistent with Fig. 3.

In order to determine which temperature-dependent chemical processes are responsible for the overall increase of net $O_x$ production with temperature, the absolute $O_x$ budgets in Fig. 4a were normalised by the total chemical loss rate of the emitted VOC (Fig. 4b). Thus Fig. 4b gives a measure of the $O_x$ production and consumption efficiency per chemical loss of VOC. The net $O_x$ production efficiency (white line in Fig. 4b) increases from $20\,°C$ to $40\,°C$ by $\sim 0.25$ molecules of $O_x$ per molecule of VOC oxidised with each $NO_x$-condition and type of isoprene emissions using the detailed MCMv3.2 chemical mechanism. A lower increase in normalised net $O_x$ production efficiency from $20\,°C$ to $40\,°C$ was obtained with the reduced chemical mechanisms ($\sim 0.2$ molecules of $O_x$ per molecule of VOC oxidised with CRIv2, CB05 and RADM2, and $\sim 0.1$ molecules of $O_x$ per molecule of VOC oxidised using MOZART-4). The increase in net $O_x$ production efficiency is due to the increased contribution with temperature of acyl peroxy radicals (ARO2) reacting with NO and the decreased net contribution with temperature of RO2NO2 (peroxy nitrates) to the normalised $O_x$ budgets.

The increased contribution of ARO2 to $O_x$ production with temperature is linked to the decreased net contribution of RO2NO2 with temperature to $O_x$ budgets as peroxy nitrates are produced from the reactions of acyl peroxy radicals with $NO_2$. The decomposition rate of peroxy nitrates is strongly temperature dependent and at higher temperatures the faster decomposition rate of $RO_2NO_2$ leads to faster release of acyl peroxy radicals and $NO_2$. Thus the equilibrium of $RO_2NO_2$ shifts towards thermal decomposition with increasing temperature leading to the increased contribution of ARO2 with temperature to $O_x$ production (Fig. 4b). The importance of peroxy nitrate decomposition to the increase of ozone with temperature has been noted by many studies, for example, Dawson et al. (2007) attributed the increase in maximum 8 h ozone mixing ratios with temperature during a modelling study over the eastern US to the decrease in PAN lifetime with temperature. Steiner et al. (2006) also recognised that the decrease in PAN lifetime with temperature contributed to the increase of ozone with temperature concluding that the combined effects of increased oxidation rates of VOC and faster PAN decomposition increased the production of ozone with temperature.

When using a temperature-independent source of isoprene emissions, the increased VOC reactivity with temperature is dominated by the increased reactivity of aldehydes at higher temperatures (up to 50 % at 40 °C), alkene and alkane emissions also have large contributions to the total VOC reactivity. The increase in VOC reactivity with temperature is primarily due to the increased emissions of isoprene with temperature in simulations using a temperature-dependent source of isoprene, aldehydes and alkanes also contribute to the total VOC reactivity when using a temperature-dependent source of isoprene. The supplementary material illustrates the contributions of different VOC functional groups to the total reactivity. The large contribution of aldehyde reactivity to total reactivity at higher temperatures is due to the increased production of aldehydes from the secondary degradation of other VOC.

As the production efficiency of $O_x$ remains constant with temperature ($\sim$ 2 molecules of $O_x$ per molecule of VOC oxidised, Fig. 4b), the rate of $O_x$ production is controlled by the oxidation rate of VOCs. Faster oxidation of VOCs with temperature speeds up the production of peroxy radicals increasing ozone production when peroxy radicals react with NO to produce $NO_2$. The reactivity of VOCs has been linked to ozone production (e.g. Kleinman (2005), Sadanaga et al. (2005)) and the review of Pusede et al. (2015) acknowledged the importance of organic reactivity and radical production to the ozone-temperature relationship. Also, the modelling study of Steiner et al. (2006) noted that the increase in initial oxidation rates of VOCs with temperature leads to increased formaldehyde concentrations and in turn an increase of ozone as formaldehyde is an important source of $HO_2$ radicals.

Our results indicate that increased VOC reactivity due to faster rate constants for the reaction with OH and the decomposition rate of peroxy nitrates are the temperature-dependent chemical processes leading to increased production of $O_x$ with temperature. Out of these two chemical processes, the increased VOC reactivity with OH with temperature had a larger influence on the increase of $O_x$ production with temperature. These results are consistent between each chemical mechanism and each $NO_x$ condition.

### 3.3 Comparison to Observations and 3D Model Simulations

The final step in our study was to compare how well our idealised box model simulations represent the real-world relationship between ozone and temperature. Firstly, we compared the box model simulations to the interpolated observations of the maximum daily 8 h mean (MDA8) of ozone from Schnell et al. (2015) and the meteorological data of the ERA-Interim re-analysis (Dee et al., 2011). Using this data set, Otero et al. (2016) showed that temperature is the main meteorological driver of ozone production during the summer (JJA) months over many regions of central Europe. A further test was to compare the box model simulations to the output from a regional 3D model as 3D models include explicit representations of transport and mixing processes which influence ozone production, and which are not well represented in our box model. We used the WRF-Chem 3D model set-up over the European domain to simulate ozone production in the year 2007 using MOZART-4 chemistry, further details are described in Mar et al. (2016).

Directly comparing the ozone mixing ratios from our idealised box model simulations to the WRF-Chem output is difficult due to significant differences in the setups of the models. For example, WRF-Chem used gas-phase and aerosol chemistry whereas our box model setup used only gas-phase chemistry, also the treatment of photolysis and VOC and $NO_x$ emissions differ between our box model and WRF-Chem. In addition to this, to include the effects of transport and mixing, the box model includes a simple mixing term representing the entrainment of clean free tropospheric air into the growing daytime boundary layer. Stagnant atmospheric conditions are characterised by low wind speeds slowing the transport of ozone and its precursors away from sources and have been correlated with high-ozone episodes in the summer over eastern US (Jacob et al., 1993). Hence, out of the metereological conditions not represented by our box model, stagnation could have the largest influence on the increase of ozone with temperature In order to investigate the sensitivity of ozone production to mixing, further box model simulations were performed without mixing approximating stagnant conditions.

Figure 5 compares the ERA-Interim reanalysis and WRF-Chem output from summer 2007 averaged over central and eastern Germany, where summertime ozone values are driven by temperature (Otero et al., 2016), to the MDA8 values of ozone from the box model simulations for each chemical mechanism with mixing (solid lines) and without mixing (dotted lines). We compare the rate of change of ozone with temperature ($m_{O3-T}$) between the box model, WRF-Chem and ERA-Interim reanalysis data. This metric has been used to quantify future ozone pollution due to the warmer temperatures predicted by climate change (Dawson et al., 2007; Rasmussen et al., 2013) and is discussed further in the review of Pusede et al. (2015). $m_{O3-T}$ is calculated as the linear slope of the increase of ozone with temperature in ppbv of ozone per °C. Polluted areas have larger $m_{O3-T}$ values than rural areas corresponding to the High-$NO_x$ and Low-$NO_x$ conditions simulated in our study. Table 3 summarises the calculated slopes of the box model simulations displayed in Fig. 5.

The linear slope of the observational data indicates an increase of 2.15 ppbv ozone per °C, this is comparable to the increase of ozone with temperature from other recent studies over urban areas: 2.2 ppbv/°C obtained over the Northeast US (Rasmussen et al., 2013) and Milan, Italy (2.8 ppbv/°C, Baertsch-Ritter et al. (2004)). Despite a high bias in simulated ozone in WRF-Chem, the rate of change of ozone with temperature from the WRF-Chem simulations (2.05 ppbv/°C) is similar to the rate of change of ozone with temperature from the observed data (2.15 ppbv/°C). The differences in ozone production between the different

chemical mechanisms with the box model are small compared to the spread of the observational and WRF-Chem data. A temperature-dependent source of isoprene with high-$NO_x$ conditions produces the highest ozone-temperature slope, but is still lower than the observed ozone-temperature slope by a factor of two. In particular, the box model simulations over-predict the ozone values at lower temperatures and under-predict the ozone values at higher temperatures compared to the observed data.

For all chemical mechanisms, the rate of increase of ozone with temperature increased in the box model simulations without mixing. The $m_{O3-T}$ calculated from the box model simulations without mixing using a temperature-dependent source of isoprene and with Maximal-$O_3$ conditions (ranging between 2.0 and 2.4 ppbv/°C) are very similar to the slopes of the observational and WRF-Chem results (2.1 and 2.2 ppbv/°C, respectively). The differences in $m_{O3-T}$ when not including mixing in the box model compared to the differences in $m_{O3-T}$ between chemical mechanisms in Table 3 show that the ozone-temperature

relationship using our box model setup is more sensitive to mixing than the choice of chemical mechanism.

    Analysis of the $O_x$ budgets, similar to that presented in Sect. 3.2, shows an increase in absolute net production of $O_x$ when simulating stagnant conditions compared to simulations including mixing (Fig. 4a). Moreover, the $O_x$ budgets normalised by the chemical loss rate of VOC for the simulations without mixing show no appreciable difference to the simulations including mixing. This analysis is displayed in the supplementary material and is consistent for each chemical mechanism and each $NO_x$

condition. Thus we conclude that the increased ozone production seen in the box model simulations with reduced mixing is due to enhanced OH reactivity from secondary VOC oxidation products.

    A slower rate of increase of ozone with temperature with our box model was obtained compared to the rate of increase of ozone with temperature of observational and 3D model simulations. The reason for this discrepancy was that the box model did not represent stagnation conditions which are relevant to real-world conditions. The lack of mixing meant that secondary VOC

oxidation products were allowed to accumulate, leading to further degradation and increased production of peroxy radicals compared with simulations including mixing. Thus the chemical processes driving the increase of ozone with temperature determined in Sect.3.2 (faster VOC oxidation and peroxy nitrate decomposition) are not altered by stagnant condition but proceed at a faster rate. ehus during stagnant conditions, stronger reductions in $NO_x$ are required to minimise the impact of increased ozone production at higher temperatures on the urban population.

**4   Conclusions**

In this study, we determined the effects of temperature on ozone production using a box model over a range of temperatures and $NO_x$ conditions with a temperature-independent and temperature-dependent source of isoprene emissions. These simulations were repeated using reduced chemical mechanism schemes (CRIv2, MOZART-4, CB05 and RADM2) typically used in 3D models and compared to the near-explicit MCMv3.2 chemical mechanism.

Each chemical mechanism produced a non-linear relationship of ozone with temperature and $NO_x$ with the most ozone produced at high temperatures and moderate emissions of $NO_x$. Conversely, lower $NO_x$ levels led to a minimal increase of ozone with temperature. Thus air quality in a future with higher temperatures would benefit from reductions in $NO_x$ emissions. Simulations with high-$NO_x$ emissions at high temperatures led to the largest differences in ozone mixing ratios predicted by

the different chemical mechanisms, future work is needed to address these inter-mechanism differences. Our results indicated that CB05 and RADM2 simulated more $NO_x$-sensitive chemistry than MCMv3.2, CRIv2 and MOZART-4. Thus for the same conditions, CB05 and RADM2 would simulate a lower increase of ozone with temperature than MCMv3.2, CRIv2 and MOZART-4 which could lead to different mitigation strategies being proposed depending on the chemical mechanism.

Faster reaction rates at higher temperatures were responsible for a greater absolute increase in ozone than increased isoprene emissions. In our simulations, ozone production was controlled by the increased rate of VOC oxidation with temperature. The net influence of peroxy nitrates increased the net production of $O_x$ per molecule of emitted VOC oxidised with temperature. Currently, chemical mechanisms do not represent the temperature-dependence of alkyl nitrate formation which may lead to discrepancies when simulating temperature-dependent ozone production over certain areas and further work assessing the
impact of this missing temperature-dependent chemical process is required.

The rate of increase of ozone with temperature using observational data over Europe was twice as high as the rate of increase of ozone with temperature when using the box model. This was consistent with our box model setup not representing stagnant atmospheric conditions that are inherently included in observational data and models including meteorology, such as WRF-Chem. In model simulations without mixing the rate of increase of ozone with temperature was faster than the simulations including mixing. The simulations without mixing and a maximal ozone production chemical regime led to very similar rates of
increase of ozone with temperature to the observational and WRF-Chem data. Furthermore, the ozone-temperature relationship was more sensitive to mixing than the choice of chemical mechanism.

*Author contributions.* T. M. Butler and J. Coates designed the experiment, J. Coates performed box model simulations and analysis. K. A. Mar and N. Ojha performed WRF-Chem model runs and provided this data. J. Coates prepared the manuscript with comments from all
co-authors.

*Acknowledgements.* The authors would like to thank Noelia Otero Felipe for assistance with the processing of the ERA-Interim data.

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

**Figure 1.** The estimated isoprene emissions (molecules isoprene cm$^{-2}$ s$^{-1}$) using MEGAN2.1 at each temperature used in the study.

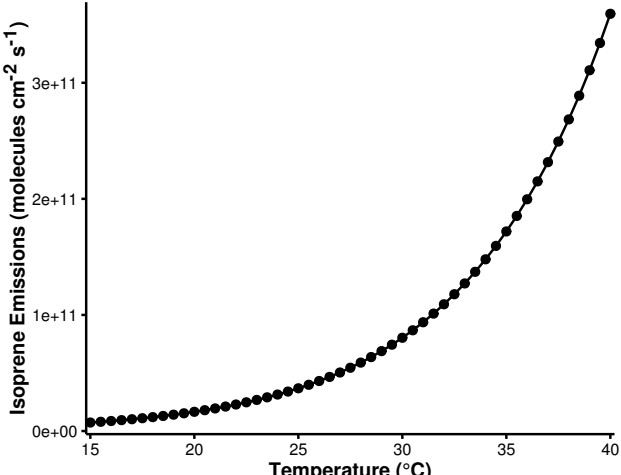

Stockwell, W. R., Kirchner, F., Kuhn, M., and Seefeld, S.: A new mechanism for regional atmospheric chemistry modeling, Journal of Geophysical Research: Atmospheres, 102, 25 847–25 879, 1997.

Vautard, R., Honoré, C., Beekmann, M., and Rouil, L.: Simulation of ozone during the August 2003 heat wave and emission control scenarios, Atmospheric Environment, 39, 2957 – 2967, 2005.

5 Vogel, B., Riemer, N., Vogel, H., and Fiedler, F.: Findings on NOy as an indicator for ozone sensitivity, Journal of Geophysical Research, 104, 3605–3620, 1999.

von Schneidemesser, E., Monks, P. S., Allan, J. D., Bruhwiler, L., Forster, P., Fowler, D., Lauer, A., Morgan, W. T., Paasonen, P., Righi, M., Sindelarova, K., and Sutton, M. A.: Chemistry and the Linkages between Air Quality and Climate Change, Chemical Reviews, pMID: 25926133, 2015.

10 von Schneidemesser, E., Coates, J., Visschedijk, A. J. H., Denier van der Gon, H. A. C., and Butler, T. M.: Variation of the NMVOC speciation in the solvent sector and the sensitivity of modelled tropospheric ozone, Atmospheric Environment, Submitted for Publication, 2016.

Wagner, P. and Kuttler, W.: Biogenic and anthropogenic isoprene in the near-surface urban atmosphere — A case study in Essen, Germany, Science of The Total Environment, 475, 104 – 115, 2014.

Wang, J., Chew, C., Chang, C.-Y., Liao, W.-C., Lung, S.-C. C., Chen, W.-N., Lee, P.-J., Lin, P.-H., and Chang, C.-C.: Biogenic isoprene in 15 subtropical urban settings and implications for air quality, Atmospheric Environment, 79, 369–379, 2013.

Yarwood, G., Rao, S., Yocke, M., and Whitten, G. Z.: Updates to the Carbon Bond Chemical Mechanism: CB05, Tech. rep., U. S Environmental Protection Agency, 2005.

**Figure 2.** Ozone mixing ratios (ppbv) as a function of total $NO_x$ emissions and temperature for each chemical mechanism using a temperature-dependent and temperature-independent source of isoprene emissions.

(a) Contours of peak ozone mixing ratios (ppbv) as a function of the total $NO_x$ emissions and temperature. The contours can be split into three separate regimes: High-$NO_x$, Maximal-$O_3$ and Low-$NO_x$ indicated in the figure.

(b) Percent difference in peak ozone from MCMv3.2 at each temperature and $NO_x$ level.

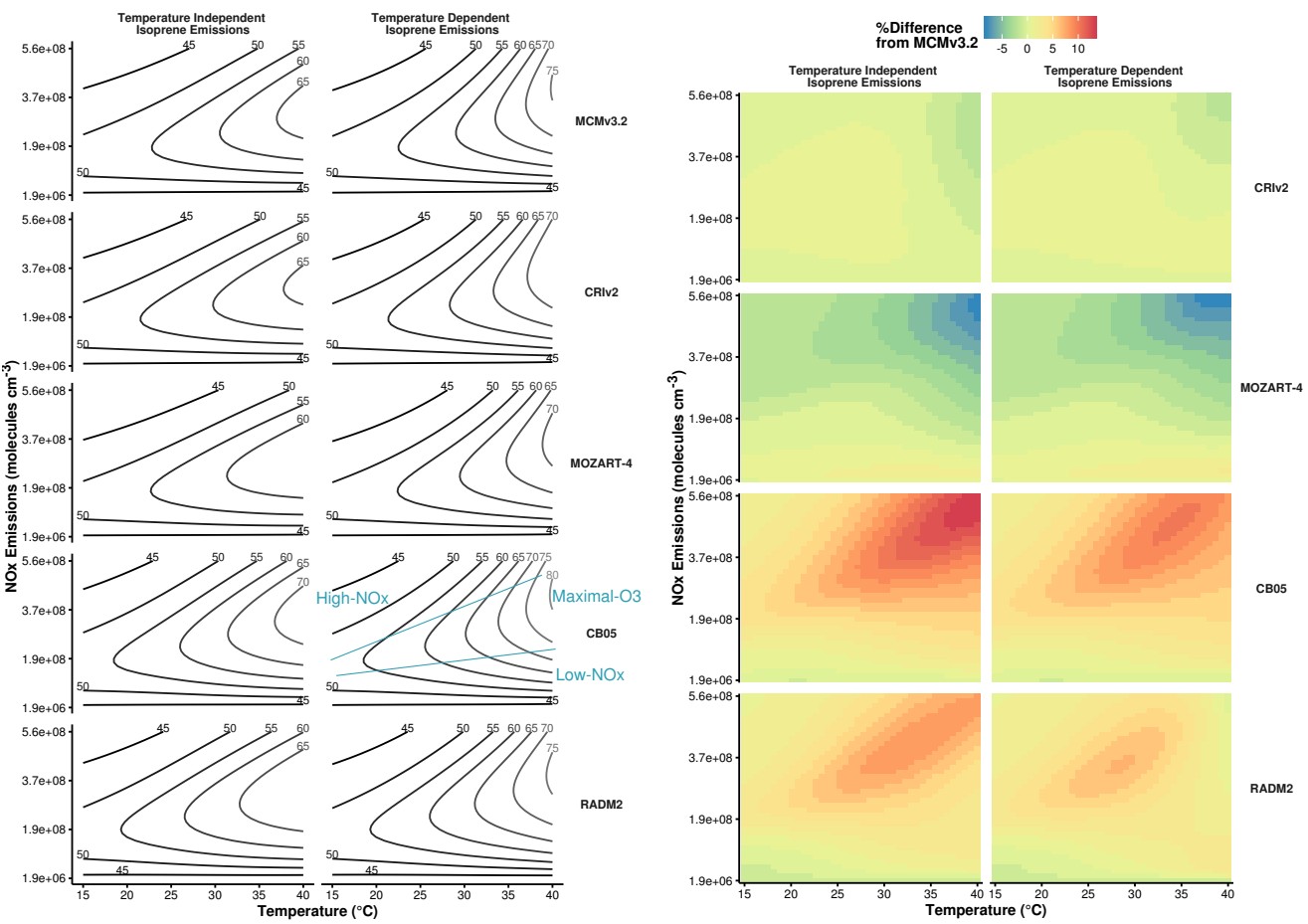

**Figure 3.** Mean ozone mixing ratios (ppbv) at each temperature after allocation to the different $NO_x$-regimes of Fig. 2a. The differences in ozone mixing ratios due to chemistry (solid line) and isoprene emissions (dotted line) are represented graphically for MOZART-4 with High-$NO_x$ conditions. Table 2 details the differences for each chemical mechanism and $NO_x$-condition.

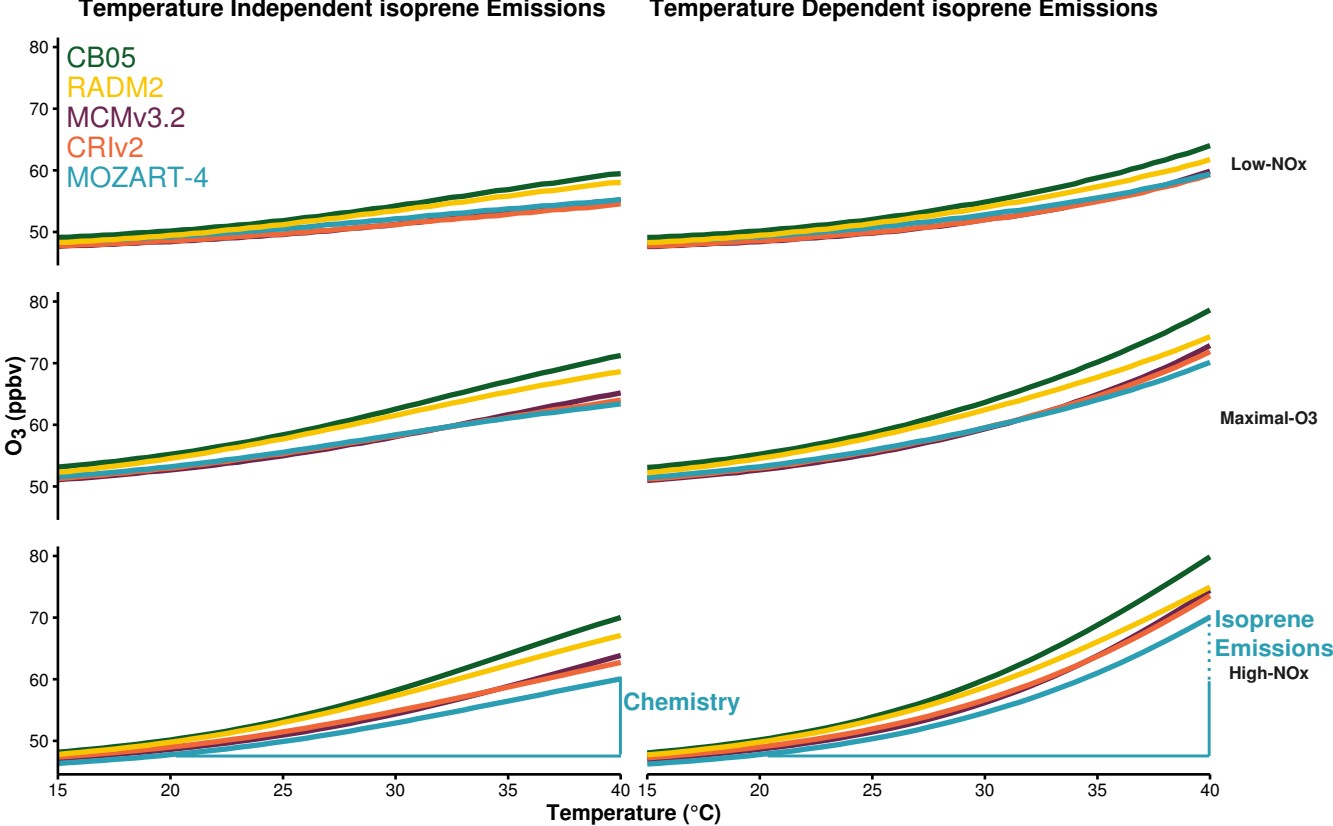

**Table 1.** Total AVOC emissions in 2011 in tonnes from each anthropogenic source category assigned from TNO-MACC_III emission inventory and temperature-independent BVOC emissions in tonnes from Benelux region assigned from EMEP. The allocation of these emissions to MCMv3.2, CRIv2, CB05, MOZART-4 and RADM2 species are found in the supplementary material.

| Source Category | Total Emissions | Source Category | Total Emissions |
|---|---|---|---|
| Public Power | 13755 | Road Transport: Diesel | 6727 |
| Residential Combustion | 21251 | Road Transport: Others | 1433 |
| Industry | 62648 | Road Transport: Evaporation | 2327 |
| Fossil Fuel | 15542 | Non-road Transport | 17158 |
| Solvent Use | 100826 | Waste | 1342 |
| Road Transport: Gasoline | 24921 | BVOC | 10702 |

**Figure 4.** Day-time production and consumption budgets of $O_x$ in the $NO_x$-regimes. The white line indicates net production or consumption of $O_x$. The net contribution of reactions to $O_x$ budgets are allocated to categories of inorganic reactions, peroxy nitrates (RO2NO2), reactions of NO with HO2, alkyl peroxy radicals (RO2) and acyl peroxy radicals (ARO2). All other reactions are allocated to the 'Other Organic' category.

(a) $O_x$ production and consumption budgets.

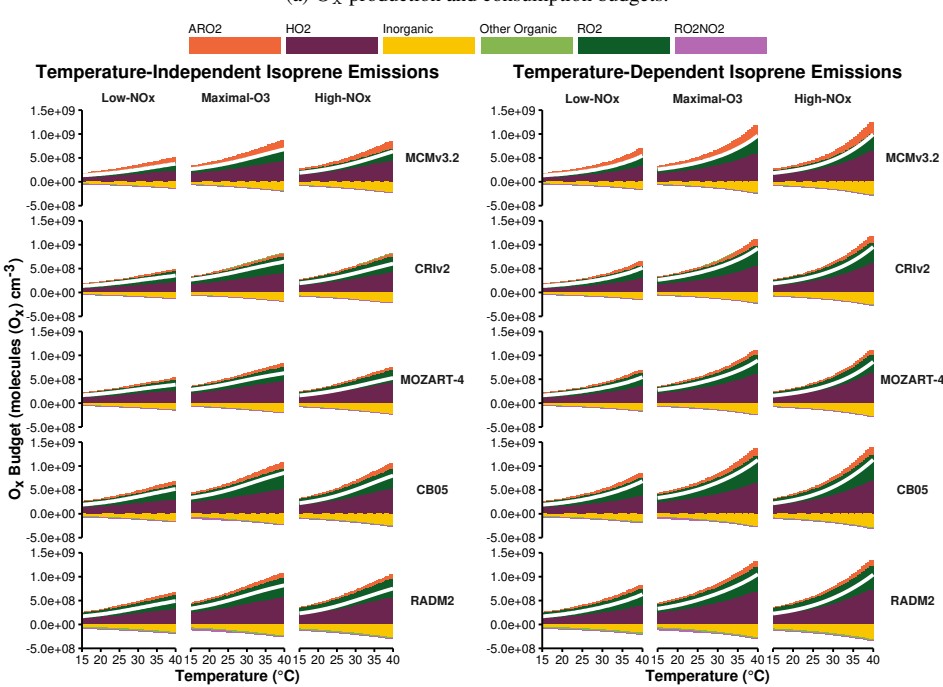

(b) $O_x$ production and consumption budgets normalised by the total loss rate of emitted VOC.

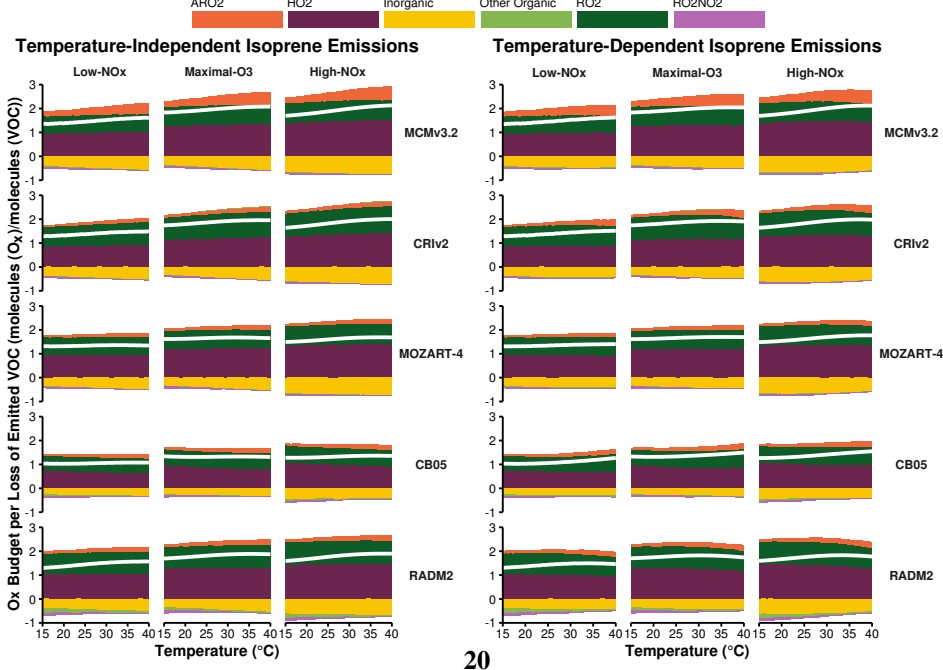

**Figure 5.** MDA8 values of ozone from the box model simulations allocated to the different $NO_x$ regimes for each chemical mechanism with mixing (solid lines) and without mixing (dashed lines). The slopes of the box model ozone-temperature correlation is compared to the summer 2007 observational data (black circles) and WRF-Chem output (purple boxes) in Table 3.

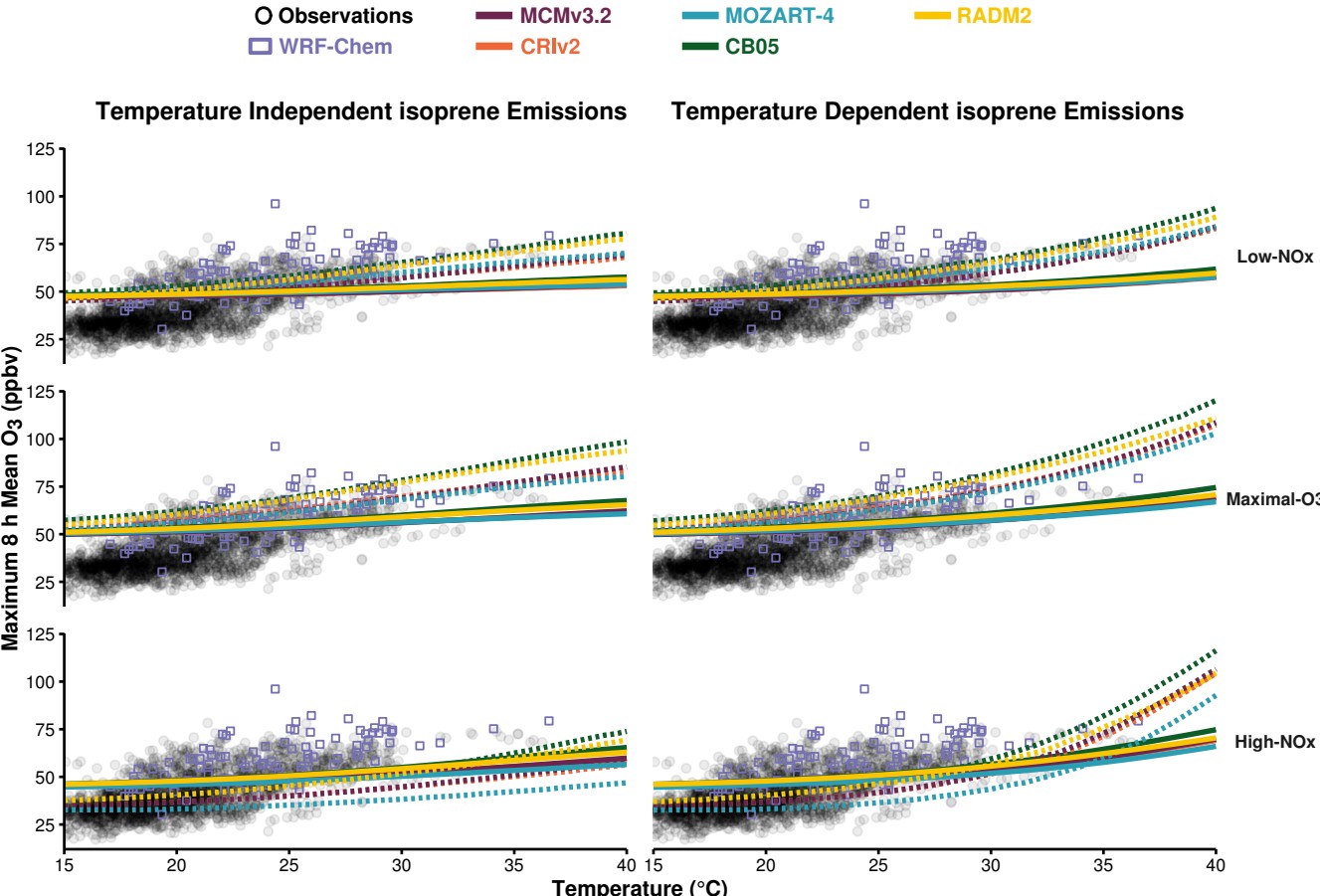

**Table 2.** Increase in mean ozone mixing ratio (ppbv) due to chemistry (i.e. faster reaction rates) and temperature-dependent isoprene emissions from 20 °C to 40 °C in the $NO_x$-regimes of Fig. 3.

| Chemical Mechanism | Source of Difference | Increase in Ozone from 20 °C to 40 °C (ppbv) | | |
|---|---|---|---|---|
| | | Low-$NO_x$ | Maximal-$O_3$ | High-$NO_x$ |
| MCMv3.2 | Isoprene Emissions | 4.6 | 7.7 | 10.6 |
| | Chemistry | 6.8 | 12.5 | 15.2 |
| CRIv2 | Isoprene Emissions | 4.8 | 7.9 | 10.8 |
| | Chemistry | 6.0 | 11.1 | 13.7 |
| MOZART-4 | Isoprene Emissions | 4.1 | 6.7 | 10.0 |
| | Chemistry | 6.0 | 10.2 | 12.3 |
| CB05 | Isoprene Emissions | 4.6 | 7.4 | 9.8 |
| | Chemistry | 9.3 | 16.0 | 19.9 |
| RADM2 | Isoprene Emissions | 3.8 | 5.7 | 7.8 |
| | Chemistry | 8.6 | 14.1 | 17.3 |

**Table 3.** Slopes ($m_{O3-T}$, ppbv per °C) of the linear fit to MDA8 values of ozone and temperature correlations in Fig. 5, indicating the increase of MDA8 in ppbv of ozone per °C. The slope of the observational data is 2.15 ppbv/°C and the slope of the WRF-Chem output is 2.05 ppbv/°C.

| Mechanism | Isoprene Emissions | Low-$NO_x$ | | Maximal-$O_3$ | | High-$NO_x$ | |
|---|---|---|---|---|---|---|---|
| | | Mixing | No Mixing | Mixing | No Mixing | Mixing | No Mixing |
| MCMv3.2 | Temperature Independent | 0.28 | 1.01 | 0.51 | 1.36 | 0.59 | 0.96 |
| | Temperature Dependent | 0.42 | 1.48 | 0.74 | 2.16 | 0.93 | 2.63 |
| CRIv2 | Temperature Independent | 0.25 | 0.93 | 0.47 | 1.27 | 0.55 | 0.88 |
| | Temperature Dependent | 0.40 | 1.44 | 0.71 | 2.09 | 0.90 | 2.52 |
| MOZART-4 | Temperature Independent | 0.25 | 0.97 | 0.44 | 1.21 | 0.49 | 0.59 |
| | Temperature Dependent | 0.38 | 1.43 | 0.65 | 1.98 | 0.81 | 2.05 |
| CB05 | Temperature Independent | 0.39 | 1.30 | 0.67 | 1.72 | 0.79 | 1.45 |
| | Temperature Dependent | 0.52 | 1.72 | 0.89 | 2.44 | 1.12 | 2.94 |
| RADM2 | Temperature Independent | 0.37 | 1.31 | 0.61 | 1.64 | 0.70 | 1.28 |
| | Temperature Dependent | 0.48 | 1.68 | 0.79 | 2.22 | 0.97 | 2.49 |