# Peer review of "The Influence of Temperature on Ozone Production under varying $NO_{\rm x}$ Conditions – a modelling study"

_Atmospheric Chemistry and Physics, 2016_

## Short Comment (SC1) · 25 May 2016

Dear colleagues, you may want to refer to Riemer et al. JGR, VOL. 104, NO. D3, PAGES, 3605-3620, FEBRUARY 20, 1999 in the revised version of your paper. Kind regards Bernhard Vogel

---

## Author Comment (AC1) · 1 Jun 2016

Thanks for the comment on our manuscript. We shall include a reference to the study Vogel et al. (1999) in the next revision of the paper.

**References**

Vogel, B. and Riemer, N. and Vogel, H. and Fiedler, F., Findings on NOy as an indicator for ozone sensitivity based on different numerical simulations, Journal of Geophysical Research: Atmospheres, 104, D3, 3605..3620, 1999.

---

## Referee Comment (RC1) · Anonymous Referee #2 · 2 Jun 2016

General comments: This paper presents a modelling study of ozone production under varying NOx conditions, concentrating on the temperature influence of the processes. Generally the paper is interesting, within the scope of ACP and should be published; however there are a few ways I feel it could be improved and these are detailed below. Generally the paper is very short. I realise that keeping things brief and to the point is sometimes a good thing and can help the reader concentrate on the salient points, however I would suggest in this case that some of the supplementary material be moved to the main text. In particular I think the model setup section would benefit from having more description in the main text rather than most of it being in the supplementary. This is important information for the paper and in this case I believe it would

assist the reader to expand the model description.

Specific comments: In section 2.1 (page 3 line 30 – page 4 line 4), several statements are made about the setup of the model that would benefit from expansion. The authors state that isoprene emissions from vegetation are the most important BVOC emissions on a global scale, however if the study was to be used for mechanisms in regional as well as global models, then could other BVOCs and other isoprene sources become important? For example in moderate to high NOx conditions of large cities could anthropogenic isoprene be important? And could monoterpene emissions (which have a potentially large effect on O3 chemistry due to their reaction rate with OH and O3 itself) also be significant? In general this seems to be a big statement to make without further discussion. The authors also state (page 4 line 3) that AVOC emissions can be effected by increased temperature due to increase evaporation but then have no further discussion as to how omitting this temperature dependence from the study may affect the results.

On page 4 line 30 it is described how isoprene emissions with varying temperature using MEGAN2.1 lead to different isoprene mixing ratios in the model, and this is then compared to isoprene measured at different temperatures during a campaign over Essen, Germany. This needs expanding. I presume MEGAN was run in the model for the particular area that the campaign took place over but this needs stating explicitly. Could the authors check their model with other campaigns that have measure isoprene (of which there are numerous worldwide in the literature)?

On page 5 line 30 a description is given that the increase in ozone due to chemistry is large than that due to increased emissions. The results are shown in figure 3 and table 2, however the paper would greatly benefit from a summary of the results in the text. On page 7 line 16, there is a paragraph describing how faster reaction of VOCs with OH with increased temperature can increase ozone production. This is backed up by references to other studies that have seen this effect. Why have the authors not included the results of their study here? Could they include some description of which

VOC + OH reactions are most dependant on temperature, which would assist readers in coming to a conclusion about which reactions and their temperature dependence should be included in any given model?

In section 3.3, a description is given of how the box model simulations in this study compare to real-world observations and the output of various 3-D models. I must admit I am a bit confused what this section is trying to say. It seems that the result is that mixing in the box model is more important to ozone formation that the choice of mechanism (which is not surprising) and I am not quite sure how any useful comparison can be made between the different mechanisms in this study and a few real world and 3-d model studies. Maybe the authors could better explain what they are trying to achieve with this section. Would a better approach be to assess what mechanisms were used in the various studies they look at and then give some steer as to whether it is the temperature dependence of the chemistry or of the emissions that is the key driver in these different cases?

Minor comments: Page 1 line 22: Could more references be added here – especially with respect to the many studies of the 2003 European heatwave ozone events? Page 3 line 13: What was 'broadly representative of urban conditions of central Europe' mean. Please be more specific with the conditions the model was run at. Page 3 line 27: The Stockwell 1990 reference seems very old. Has there been more recent advances in the knowledge of ozone production chemistry that might make this obsolete? Page 8 line 25: The authors should consider showing the actual production and consumption budgets in the main text rather than the supplementary.

---

## Referee Comment (RC2) · Anonymous Referee #1 · 15 Jul 2016

This manuscript presents a modeling study of aspects of the dependence of ozone production on temperature as a function of NOx level. Ozone chemistry is tested for two temperature-dependent effects: temperature-driven isoprene emissions and temperature-dependent reaction rates. Results are computed separately for different NOx regimes, compared across multiple mechanisms, and compared with maximum daily 8-h mean (MDA8) surface observations and simulated MDA8 surface ozone using a 3D model (WRF-CHEM). The paper addresses an interesting question, but requires revision prior to publication.

Generally, the paper needs clearer focus and to supply more model details/results/discussion pertinent to this focus. While the stated goal is to assess the

temperature dependence of ozone chemistry as a function of NOx, at the end, I am not exactly sure what new has been learned. Really, what I think is missing is just a dedicated discussion section, the 'Results' read like a true results section, rather than combined results and discussion, so it is difficult for the reader to understand the significance of the calculations.

Section 3.1: The reasons behind the ozone impacts from temperature-dependent re-action rates are unclear because it is not stated explicitly what this term includes. If I understand correctly, only temperature-dependent reaction rates, k(T), are being tested (page 4, lines 1–6). Generally, the model description does not give the reader sufficient information to understand what causes the changes reported in Table 2. For example, the text states the RONO2 formation is temperature dependent, at least in some mech-anisms, but does this refer to the RONO2 branching ratio?

It is interesting to see how the five different mechanisms capture these effects; how-ever, there is little discussion of what is learned about the different mechanisms by testing them in this way. Can the last paragraph of Sect. 3.1 be expanded? Also, I have difficulty discerning differences between panels in Fig. 2. Is there a way to high-light key differences here? Regarding Fig. 3, something has been lost in translation - there are some floating numbers in the upper left corner of the top-left panel, but the different colors are not labeled, either in the text or caption. I deduce that green is CB05, purple is MCM3.2, blue is MOZART-4, and orange and yellow are each either RADM2 or CRIv2. I would be interested to read more, not just about what causes the differences between the curves, but also the implications for studying air quality and chemistry. Finally, the NOx regime distinction is derived from each individual model's H2O2:HNO3. Why not simply use the shape of simulated PO3 versus NOx. A missing piece of information is whether maximal O3 - at different temperatures - occurs at the same NOx level in each mechanism. Differences in the NOx level of maximal O3 for different mechanisms have consequences for air quality decision-making.

Section 3.2: Provide a statement as to why the ozone production and consumption

budget is informative for understanding the temperature dependence of ozone, i.e. what is gained compared to thinking about production alone. Also, can an equation be provided for the production and consumption budgeting? This section is in need of discussion and summary. There are many panels in Fig. 4 and it is not obvious to me what the take-away point(s) are.

Section 3.3: Before the authors talk about mixing, the WRF-CHEM and box MOZART-4 results should be compared directly and discussed. The importance of atmospheric mixing appears for the first time in Section 3.3, at which time, the paper states it is the most important term in mO3-T. At this stage in the manuscript, I am left wondering what is this paper actually about. How does Section 3.3 relate to the previous two sections? A subsequent discussion would be helpful.

Minor comments:

More information should be provided in the introduction. The three sentences in the paper's first paragraph do not really follow logically. I am not familiar with the Otero paper and this single-sentence description does not stand on its own–temperature was shown to be a driver of which process?

Fig. 2: The ozone contours are labeled left to right: 5, 50, 55, 0, 5, 0, 5. The y-axis reads: 10, 10, 30, 50. On the x-axis, the 4 of 40 has been lost.

Fig. 5: The majority of measured O3 data are found at lower temperatures, so fitting the calculated O3 with a straight line across the whole temperature range may not be representative.

Fig. 5: y-axis reads 25, 50, 5, 100, 125; on the x-axis, the 4 of 40 has been lost.

---

## Author Response (AR1)

**Response to Referee #1**

We would like to thank the reviewer for the review that has enabled us to improve our manuscript; our responses to the review points and comments are found below.

**Review Point 1:** Generally, the paper needs clearer focus and to supply more model details/results/discussion pertinent to this focus. While the stated goal is to assess the temperature dependence of ozone chemistry as a function of NOx, at the end, I am not exactly sure what new has been learned. Really, what I think is missing is just a dedicated discussion section, the 'Results' read like a true results section, rather than combined results and discussion, so it is difficult for the reader to understand the significance of the calculations.

**Author Response:** We thank the reviewer for communicating these concerns. In order to address these concerns, we have updated each subsection in the 'Results and Discussion' section to include further discussion. The changes to the manuscript are detailed in the responses to the review comments on each subsection below.

**Review Point 2:** The reasons behind the ozone impacts from temperature-dependent reaction rates are unclear because it is not stated explicitly what this term includes. If I understand correctly, only temperature-dependent reaction rates, k(T), are being tested (page 4, lines 16). Generally, the model description does not give the reader sufficient information to understand what causes the changes reported in Table 2. For example, the text states the RONO2 formation is temperature dependent, at least in some mechanisms, but does this refer to the RONO2 branching ratio?

**Author Response:** The reviewer is correct in saying that only temperature-dependent reaction rates, k(T), are tested in our study. In Sect. 2.1 (Page 4, Line 2), the temperature-dependent processes included in the chemical mechanisms are presented. We have clarified that the branching ratio of $RONO_2$ formation is not represented as a temperature-dependent process by any chemical mechanism by adding the following text to the manuscript at Page 4, Line 6: *Furthermore, none of the chemical mechanisms in our study represent the $RONO_2$ branching ratio as a temperature dependent process. Laboratory experiments have shown the temperature dependence of the $RONO_2$ branching ratio for some VOCs (Atkinson et al., 1987) but generally $RONO_2$ chemistry is not well known (Pusede et al., 2015) and this level of detail is not represented by the chemical mechanisms. Before chemical mechanisms can include the temperature-dependence of the $RONO_2$ branching ratio, further research is required.*

**Review Point 3:** It is interesting to see how the five different mechanisms capture these effects; however, there is little discussion of what is learned about the different mechanisms by testing them in this way. Can the last paragraph of Sect. 3.1 be expanded? Also, I have difficulty discerning differences between panels in Fig. 2. Is there a way to highlight key differences here? Regarding Fig. 3, something has been lost in translation - there are some floating numbers in the upper left corner of the top-left panel, but the different colors are not labeled, either in the text or caption. I deduce that green is CB05, purple is MCM3.2, blue is MOZART-4, and orange and yellow are each either

RADM2 or CRIv2. I would be interested to read more, not just about what causes the differences between the curves, but also the implications for studying air quality and chemistry. Finally, the NOx regime distinction is derived from each individual model's H2O2:HNO3. Why not simply use the shape of simulated PO3 versus NOx. A missing piece of information is whether maximal O3 - at different temperatures - occurs at the same NOx level in each mechanism. Differences in the NOx level of maximal O3 for different mechanisms have consequences for air quality decision-making.

**Author Response:** We thank the reviewer for the comments on Sect. 3.1 of the manuscript and have made changes to the manuscript based on these points. Figure 2 has been updated with a subfigure (Fig. 2b) displaying the relative differences in ozone mixing ratios in each chemical mechanism at each temperature, $NO_x$ condition from the MCMv3.2 when using a temperature-independent and temperature-dependent source of isoprene emissions. Based on this figure, a discussion of the differences between the chemical mechanisms is included in the manuscript (Page 5, Line 26): *As shown in Fig. 2b, regions of high temperatures and high $NO_x$ emissions generally lead to the largest inter-mechanism differences between ozone mixing ratios using reduced chemical mechanisms from the MCMv3.2 (up to $13$ %). These differences in peak ozone mixing ratio produced from the reduced chemical mechanisms compared with the MCMv3.2 in each $NO_x$ condition are consistent with Fig. 3 (described below) where RADM2 and CB05 generally produced higher ozone levels than the MCMv3.2. Also consistent with Fig. 3, CRIv2 produced the most similar amounts of ozone to the MCMv3.2 in each $NO_x$ condition whereas MOZART-4 tended to produce lower ozone mixing ratios than the MCMv3.2 in High-$NO_x$ conditions. In Fig. 3, a maximum difference of $10$ ppbv between ozone mixing ratios produced using the chemical mechanisms is reached at $40\ ^\circ C$ in the High-$NO_x$ state when using both a temperature-independent and temperature-dependent source of isoprene emissions.*

The issue with the display of the figures appears to be an issue with the default pdfviewer of the Chrome browser, downloading the file from the default pdfviewer in Chrome does not correct this issue. Opening the ACPD pdf using the Firefox browser and a pdfviewer extension of the Chrome browser (Kami) solves the issue and the manuscript is displayed as intended by the authors. We shall ensure that the updated manuscript does display correctly with the default pdfviewer in Chrome.

We used the H2O2:HNO3 ratio to assign a NOx regime as this has been used previously in other studies for this purpose, we have updated the manuscript to include examples. Sect. 3.1 (Page 6, Line 1) was updated: *This ratio was used by Sillman (1995) and Staffelbach et al. (1997) to designate ozone to $NO_x$ regimes based on $NO_x$ and VOC levels.*

The manuscript (Page 6, Line 17) was also updated to include further discussion on the $NO_x$ emission required for the Maximal-O3 regime with each chemical mechanism, to aid this discussion a new figure was generated and included in the updated supplementary material: *The $NO_x$ emissions required for maximum ozone production (the contour ridges in Fig. 2a) at each temperature is displayed in Fig. 1 of the supplementary material. This figure illustrates that RADM2 and CB05 require higher $NO_x$ emissions than the MCMv3.2 to achieve maximum ozone production at each temperature for*

*both a temperature-independent and temperature-dependent source of isoprene emissions. At 20 °C, maximum ozone production is reached with ∼ 30 % more $NO_x$ emissions using CB05 and RADM2 than the MCMv3.2 with a temperature-independent and temperature-dependent source of isoprene emissions. The CRIv2 and MOZART-4 chemical mechanisms require very similar $NO_x$ emissions to the MCMv3.2 at each temperature to produce maximum levels of ozone. Thus when modelling the air quality over a particular region using RADM2 and CB05, these mechanisms would be expected to simulate more $NO_x$-sensitive chemistry than the MCMv3.2, CRIv2 and MOZART-4 chemical mechanisms for the same conditions (i.e. emissions, meteorology and radiation). These differences in the ozone production regime using different chemical mechanisms highlight the need for air quality studies to assess the chemical scheme used by the model, otherwise differing mitigation strategies may be proposed.*

**Review Point 4:** Provide a statement as to why the ozone production and consumption budget is informative for understanding the temperature dependence of ozone, i.e. what is gained compared to thinking about production alone. Also, can an equation be provided for the production and consumption budgeting? This section is in need of discussion and summary. There are many panels in Fig. 4 and it is not obvious to me what the take-away point(s) are.

**Author Response:** We agree with these review points and have provided additional information about the analysis of ozone production and consumption budgets, including equations and further description of how Fig. 4 was obtained. Section 3.2 of the manuscript (Page 7, Line 1) was updated: *Since chemical reactions contributing to both production and consumption of $O_x$ ($\equiv O_3 + NO_2 + O(^1D) + O$) have temperature-dependent rate constants, we analysed the production and consumption budgets of $O_x$ to determine the temperature-dependent chemical processes controlling the increase of ozone with temperature which was shown in Fig. 3. The $O_x$ budgets displayed in Fig. 4 are assigned to each $NO_x$ regime for each chemical mechanism and source of isoprene emissions. The net production or consumption of $O_x$ is also indicated in Fig. 4.*

*Figure 4 was obtained by determining the chemical reactions producing and consuming $O_x$ and then allocating these reactions to important categories. Reactions of peroxy radicals with NO produce $O_x$ and the peroxy radicals are divided into 'HO2', 'RO2', 'ARO2' categories representing the reactions of NO with $HO_2$, alkyl peroxy radicals and acyl peroxy radicals respectively. Thus at each time step the $O_x$ production rate is given by*

$$k_{HO_2+NO}[HO_2][NO] + \sum_i k_{RO_{2,i}+NO}[RO_{2,i}][NO] + \sum_j k_{ARO_{2,j}+NO}[ARO_{2,j}][NO] \qquad (1)$$

*for each alkyl peroxy radical i and acyl peroxy radical j. The net contributions of peroxy nitrates, inorganic reactions and any other remaining organic reactions to the $O_x$ budget are represented by the 'RO2NO2', 'Inorganic' and 'Other Organic' categories in Fig. 4. The net contributions of these categories to the $O_x$ budget was calculated by subtracting the consumption rate from the production rate of the reactions contributing to each category. For example, peroxy nitrates produce $O_x$ when*

thermally decomposing or reacting with OH and consume $O_x$ when produced. Hence, at each time step the net contribution of RO2NO2 to the $O_x$ budget was calculated by

$$\sum_k k_{RO_2NO_{2,k}}[RO_2NO_{2,k}] + \sum_k k_{RO_2NO_{2,k}+OH}[RO_2NO_{2,k}][OH] - \sum_k k_{RO_{2,k}+NO_2}[RO_{2,k}][NO_2]$$
(2)

for each peroxy nitrate species $k$. The cumulative day-time budgets were calculated by summing the net contributions of the reaction rates of each category over the day-time period. The ratio of net ozone to net $O_x$ production was practically constant with temperature in all cases showing that using $O_x$ budgets as a proxy for ozone budgets was suitable at each temperature in our study.

The main results of the section are summarised and discussed in the final paragraph of Sect. 3.2 (Page 8) of the updated manuscript: *Our results indicate that increased VOC reactivity due to faster rate constants for the reaction with OH and the decomposition rate of peroxy nitrates are the temperature-dependent chemical processes leading to increased production of $O_x$ with temperature. Out of these two chemical processes, the increased VOC reactivity with OH with temperature had a larger influence on the increase of $O_x$ production with temperature. These results are consistent between each chemical mechanism and each $NO_x$ condition.*

**Review Point 5:** Before the authors talk about mixing, the WRF-CHEM and box MOZART-4 results should be compared directly and discussed. The importance of atmospheric mixing appears for the first time in Section 3.3, at which time, the paper states it is the most important term in mO3-T. At this stage in the manuscript, I am left wondering what is this paper actually about. How does Section 3.3 relate to the previous two sections? A subsequent discussion would be helpful.

**Author Response:** We thank the reviewer for these comments on Sect. 3.3, we have updated the introductory part of this section to include further details of the motivations of the analysis and also discuss the use of the metric of $m_{O3-T}$ for comparing the box model, observations and WRF-Chem output. Furthermore the box model simulations looking at the effect of mixing on the increase of ozone with temperature are discussed in more detail. Accordingly, Section 3.3 (Page 8, Line 5) of the manuscript was updated: *The final step in our study was to compare how well our idealised box model simulations represent the real-world relationship between ozone and temperature. Firstly, we compared the box model simulations to the interpolated observations of the maximum daily 8 h mean (MDA8) of ozone from Schnell et al. (2015) and the meteorological data of the ERA-Interim re-analysis (Dee et al., 2011). Using this data set, Otero et al. (2016) showed that temperature is the main meteorological driver of ozone production during the summer (JJA) months over many regions of central Europe. A further test was to compare the box model simulations to the output from a regional 3D model as 3D models include explicit representations of transport and mixing processes which influence ozone production, and which are not well represented in our box model. We used the WRF-Chem 3D model set-up over the European domain to simulate ozone production in the year 2007 using MOZART-4 chemistry, further details are described in Mar et al. (2016).*

[revised manuscript text omitted]

**Minor Comments 1:** More information should be provided in the introduction. The three sentences in the papers first paragraph do not really follow logically. I am not familiar with the Otero paper and this single-sentence description does not stand on its own temperature was shown to be a driver of which process?

**Author Response:** We thank the reviewer for this comment and have updated the introduction (Page 1, Line 21): *In particular, heatwaves, characterised by high temperatures and stagnant meteorological conditions, are correlated with high ozone levels as was the case during the European heatwave in 2003 (Solberg et al., 2008; Vautard et al., 2005).*

**Minor Comments 2:** Fig. 2: The ozone contours are labeled left to right: 5, 50, 55, 0, 5, 0, 5. The y-axis reads: 10, 10, 30, 50. On the x-axis, the 4 of 40 has been lost.

**Author Response:** As discussed in the answer to Review Point 3, the issue with figures appears to be a problem with the default pdfviewer of the Chrome browser. We shall ensure that the updated manuscript does display correctly with the default pdfviewer in Chrome.

**Minor Comments 3:** Fig. 5: The majority of measured O3 data are found at lower temperatures, so fitting the calculated O3 with a straight line across the whole temperature range may not be representative.

**Author Response:** We agree with the reviewer that a linear regression may not be the best choice for the observation. However, as a metric $m_{O3\text{-}T}$, defined as the linear slope of the increase of ozone with temperature, was the most appropriate with which to compare the observations not only to other studies but also to the box model and WRF-Chem output. The manuscipt was updated as outlined in the response to Review Point 5 and Page 8, Line 14 was also updated to compare $m_{O3\text{-}T}$ of the observational data to that from other studies: *The linear slope of the observational data indicates an increase of 2.15 ppbv ozone per $^{\circ}C$, this is comparable to the increase of ozone with temperature from other recent studies over urban areas: 2.2 ppbv/$^{\circ}C$ obtained over the Northeast US (Rasmussen et al., 2013) and Milan, Italy (2.8 ppbv/$^{\circ}C$, Baertsch-Ritter et al. (2004)).*

**Minor Comments 4:** Fig. 5: y-axis reads 25, 50, 5, 100, 125; on the x-axis, the 4 of 40 has been lost.

**Author Response:** See response to Minor Comments 2.

**Response to Referee #2**

We would like to thank the reviewer, we feel that this review has enabled us to improve our manuscript; our responses to the review points and comments are found below.

**Review Point 1:** Generally the paper is very short. I realise that keeping things brief and to the point is sometimes a good thing and can help the reader concentrate on the salient points, however I would suggest in this case that some of the supplementary material be moved to the main text. In particular I think the model setup section would benefit from having more description in the main text rather than most of it being in the supplementary. This is important information for the paper and in this case I believe it would assist the reader to expand the model description.

**Author Response:** We agree that a more detailed methodology description is beneficial to the paper and this was addressed in the published ACPD paper. The supplementary material published in ACPD includes only the information related to the speciated VOC emissions used in each chemical mechanism and no other information pertaining to the model setup. We believe that the reviewer may be commenting on a previous uploaded version, the only difference between the published ACPD paper and this previous version is the description of the model setup that was previously included in the supplement. Thus all other review comments be the reviewer are valid and addressed below.

Furthermore, the methodology section has also been updated in reponse to Specific Comments #1 and #2 below and Review Point #2 from the first referee.

**Specific Comments 1:** In section 2.1 (page 3 line 30 page 4 line 4), several statements are made about the setup of the model that would benefit from expansion. The authors state that isoprene emissions from vegetation are the most important BVOC emissions on a global scale, however if the study was to be used for mechanisms in regional as well as global models, then could other BVOCs and other isoprene sources become important? For example in moderate to high NOx conditions of large cities could anthropogenic isoprene be important? And could monoterpene emissions (which have a potentially large effect on O3 chemistry due to their reaction rate with OH and O3 itself) also be significant? In general this seems to be a big statement to make without further discussion. The authors also state (page 4 line 3) that AVOC emissions can be effected by increased temperature due to increase evaporation but then have no further discussion as to how omitting this temperature dependence from the study may affect the results.

**Author Response:** We agree that further discussion of these statements is required. In order to respond to this review point and also Review Point #1, we have updated the manuscript (Sect. 2.1, Page 4, Line 7) with the following text: *Many types of VOCs are emitted from vegetation with isoprene and monoterpenes globally having the largest emissions, 535 and 162 Tg yr$^{-1}$ respectively (Guenther et al., 2012). Temperature-dependent emissions of these highly-reactive BVOC in urban areas during the summer months have been linked to high levels of ozone pollution. For example, Wang et al. (2013) attributed high summertime levels of ozone in Taipei to increased isoprene emissions from vegetation during the hotter summer months. Vegetation in urban areas also provides additional*

ozone sinks through stomatal uptake and ozonolysis of emitted BVOCs, the review of Calfapietra et al. (2013) discusses the role of BVOCs emitted by trees in urban areas in more detail.

Biogenic emissions of monoterpenes and isoprene are included in all model simulations. Model runs using a temperature-dependent source of BVOC emissions considered only the temperature-dependence of isoprene emissions as specified by MEGAN2.1 (Guenther et al., 2012), Sect. 2.3 provides further details. Since isoprene is the most important BVOC on the global scale, we focused on the influence of the temperature-dependent biogenic emissions of isoprene on ozone levels. Future work should assess the influence of temperature-dependent biogenic emissions of monoterpenes on ozone production. In the temperature-dependent set of model simulations, only isoprene emissions were dependent on temperature and all other emissions were constant in all simulations. In reality, evaporative emissions from anthropogenic sources increase with temperature (Rubin et al., 2006) and isoprene has also been measured from vehicular exhausts (Borbon et al., 2001). Representing a temperature-dependent evaporative source of AVOC and an anthropogenic source of isoprene requires detailed local knowledge of these emission sources (such as the traffic fleet). Since our box modelling study was designed as an idealised study and not to characterise the influence of all temperature-dependent emission sources in a particular region, we have not considered the potentially larger increase of ozone at higher temperatures due to these additional emission sources. Further modelling work assessing the influence of these temperature-dependent emission sources on ozone production would be useful for mitigating ozone pollution in urban areas.

**Specific Comments 2:** On page 4 line 30 it is described how isoprene emissions with varying temperature using MEGAN2.1 lead to different isoprene mixing ratios in the model, and this is then compared to isoprene measured at different temperatures during a campaign over Essen, Germany. This needs expanding. I presume MEGAN was run in the model for the particular area that the campaign took place over but this needs stating explicitly. Could the authors check their model with other campaigns that have measure isoprene (of which there are numerous worldwide in the literature)?

**Author Response:** Our idealised study was not designed to represent the biogenic emissions of isoprene over a particular region. For this reason we did not run MEGAN over a particular area as this requires detailed knowledge of the specific vegetation. The aim of the study was to determine the additional influence of temperature-dependent isoprene emissions on top of the temperature-dependent chemistry used by different chemical mechanisms. Thus the study was designed to produce similar isoprene mixing ratios at a reference temperature (we chose 20 °C) and using MEGAN to specify the temperature-dependent profile of isoprene emissions. We have updated the manuscript to further clarify our reasoning in Page 5, Line 4: *The aim of the study was to determine the additional influence of temperature-dependent isoprene emissions on top of the temperature-dependent chemistry. In order to achieve this aim, we chose the MEGAN2.1 parameters used to calculate isoprene emissions online by the model to give similar isoprene mixing ratios at 20 °C to the temperature-independent emissions of isoprene. MEGAN2.1 was used to reflect the*

*temperature-dependent emission profile of isoprene emissions and not to accurately represent the isoprene emissions of a particular region.*

**Specific Comments 3:** On page 5 line 30 a description is given that the increase in ozone due to chemistry is large than that due to increased emissions. The results are shown in figure 3 and table 2, however the paper would greatly benefit from a summary of the results in the text.

**Author Response:** We agree with the reviewer and have updated the manuscript (Page 6, Line 31) to include a summary of this section: *Our simulations produced a non-linear relationship between ozone, temperature and $NO_x$ with the absolute increase in ozone with temperature due to temperature-dependent chemistry larger than the increase in ozone with temperature due to temperature-dependent isoprene emissions. These results are consistent between each chemical mechanism, although for the same $NO_x$ and VOC conditions RADM2 and CB05 simulate a more $NO_x$-sensitive regime at the same temperature than the other chemical mechanisms (MCMv3.2, CRIv2, MOZART-4).*

**Specific Comments 4:** On page 7 line 16, there is a paragraph describing how faster reaction of VOCs with OH with increased temperature can increase ozone production. This is backed up by references to other studies that have seen this effect. Why have the authors not included the results of their study here? Could they include some description of which VOC + OH reactions are most dependant on temperature, which would assist readers in coming to a conclusion about which reactions and their temperature dependence should be included in any given model?

**Author Response:** We agree with the reviewer and have included further information (Page 8, Line 4) about which VOC are most important for the increase of ozone with temperature: *When using a temperature-independent source of isoprene emissions, the increased VOC reactivity with temperature is dominated by the increased reactivity of aldehydes at higher temperatures (up to 50 % at 40 °C), alkene and alkane emissions also have large contributions to the total VOC reactivity. The increase in VOC reactivity with temperature is primarily due to the increased emissions of isoprene with temperature in simulations using a temperature-dependent source of isoprene, aldehydes and alkanes also contribute to the total VOC reactivity when using a temperature-dependent source of isoprene. The supplementary material illustrates the contributions of different VOC functional groups to the total reactivity. The large contribution of aldehyde reactivity to total reactivity at higher temperatures is due to the increased production of aldehydes during the secondary degradation of other VOC.*

**Specific Comments 5:** In section 3.3, a description is given of how the box model simulations in this study compare to real-world observations and the output of various 3-D models. I must admit I am a bit confused what this section is trying to say. It seems that the result is that mixing in the box model is more important to ozone formation that the choice of mechanism (which is not surprising) and I am not quite sure how any useful comparison can be made between the different mechanisms in this study and a few real world and 3-d model studies. Maybe the authors could better explain what they are trying to achieve with this section. Would a better approach be to

assess what mechanisms were used in the various studies they look at and then give some steer as to whether it is the temperature dependence of the chemistry or of the emissions that is the key driver in these different cases?

**Author Response:** Based on the comments of Reviewer #1 as well as this review point, we have updated Section 3.3 to include more details of the motivation of this comparison and a more appropriate discussion of the results. The changes to the manuscript are found in the response to Review Point 5 of Reviewer #1.

**Minor Comments:**

**Minor Comments 1:** Page 1 line 22: Could more references be added here especially with respect to the many studies of the 2003 European heatwave ozone events?

**Author Response:** We agree and have included more references in the updated manuscript: *In particular, heatwaves, characterised by high temperatures and stagnant meteorological conditions, are correlated with high ozone levels as was the case during the European heatwave in 2003 (Solberg et al., 2008; Vautard et al., 2005).*

**Minor Comments 2:** Page 3 line 13: What was broadly representative of urban conditions of central Europe mean. Please be more specific with the conditions the model was run at.

**Author Response:** We have updated the manuscript to provide further information (Page 3, Line 14): *Our simulations were designed were designed as an idealised case and not to be exact representations of any particular place. The simulations used a latitude of 51 °N, broadly representative of conditions in central Europe, and were run for daylight hours in one full day.*

**Minor Comments 3:** Page 3 line 27: The Stockwell 1990 reference seems very old. Has there been more recent advances in the knowledge of ozone production chemistry that might make this obsolete?

**Author Response:** We agree with the reviewer that RADM2 is indeed an old reference, however this chemical mechanism is still used by many modelling groups. We have clarified this further in the manuscript (Page 3, Line 34): *These reduced chemical mechanisms were chosen as they are all currently used by modelling groups in 3D regional and global models (Baklanov et al., 2014).*

**Minor Comments 4:** Page 8 line 25: The authors should consider showing the actual production and consumption budgets in the main text rather than the supplementary.

**Author Response:** We agree with the reviewer and have updated Fig. 4 to show the absolute production and consumption budgets as well as the normalised production and consumption budgets. The manuscript was updated to reflect this and details are found in the response to Review Point 4 of Reviewer #1.

**References**

[revised manuscript text omitted]